# The Role of Exosomes in the Crosstalk between Adipocytes and Liver Cancer Cells

**DOI:** 10.3390/cells9091988

**Published:** 2020-08-29

**Authors:** Leslimar Rios-Colon, Elena Arthur, Suryakant Niture, Qi Qi, John T. Moore, Deepak Kumar

**Affiliations:** Julius L. Chambers Biomedical Biotechnology Research Institute, North Carolina Central University, Durham, NC 27707, USA; lrioscolon@nccu.edu (L.R.-C.); earthur1@nccu.edu (E.A.); sniture@nccu.edu (S.N.); qqi@nccu.edu (Q.Q.); jmoor208@NCCU.EDU (J.T.M.)

**Keywords:** hepatocellular carcinoma, obesity, adipocytes, exosomes, adipocyte exosomes, exosomal cargo, cell signaling

## Abstract

Exosomes are membrane-bound extracellular vesicles (EVs) that transport bioactive materials between cells and organs. The cargo delivered by exosomes can alter a wide range of cellular responses in recipient cells and play an important pathophysiological role in human cancers. In hepatocellular carcinoma (HCC), for example, adipocyte- and tumor-secreted factors contained in exosomes contribute to the creation of a chronic inflammatory state, which contributes to disease progression. The exosome-mediated crosstalk between adipocytes and liver cancer cells is a key aspect of a dynamic tumor microenvironment. In this review, we summarize the role of increased adiposity and the role of adipocyte-derived exosomes (AdExos) and HCC-derived exosomes (HCCExos) in the modulation of HCC progression. We also discuss recent advances regarding how malignant cells interact with the surrounding adipose tissue and employ exosomes to promote a more aggressive phenotype.

## 1. Introduction

Hepatocellular carcinoma (HCC) is the most common type of liver cancer and one of the most commonly occurring cancers in both men and women worldwide [1]. Men are most affected by this disease, with three times the probability of being diagnosed compared to women [1]. According to the American Cancer Society (ACS), liver cancer will account for over 42,000 estimated new cases and 30,000 cancer deaths in the United States in 2020. The relative 5-year survival rate for this type of cancer is approximately 20%, and is highly dependent on the stage of diagnosis (American Cancer Society, Facts and Figures 2020, American Cancer Society, Atlanta, Ga. 2020). The incidence of HCC is rapidly increasing, compared to any other cancer in the United States, as a result of modifiable behaviors such as excess nutrition, increased alcohol consumption, smoking, and chronic infection with hepatitis B virus (HBV) or hepatitis C virus (HCV) [2].

Mounting evidence suggests that there is a strong link between adipose tissue, inflammation, non-alcoholic fatty liver disease (NAFLD) and alcoholic fatty liver disease (AFLD), and the risk of HCC occurrence [3,4,5,6,7]. Abdominal obesity, high body weight and diets enriched in fat trigger the onset and progression of NAFLD [8,9], which may ultimately lead to fibrosis, cirrhosis and HCC. Heavy consumption of alcohol is the major risk factor involved in AFLD, and consumption of alcohol promotes the development of HCC via direct genotoxic mechanisms or indirectly by inducing liver cirrhosis [8,9]. The link between increased adiposity and liver diseases, such as NAFLD and AFLD, is mediated in part by an increased low-grade inflammatory state as a result of factors secreted by adipose tissue [10]. To understand the role of such factors in the crosstalk between adipose tissue and the liver, we will first discuss adipogenesis and the role of obesity in the development of HCC.

## 2. Adipogenesis and Adipocytes

Among the main functions of the adipose tissue is the storage and release of lipids to maintain energy homeostasis [11]. Adipose tissue can be divided into three different types: white adipose tissue (WAT), brown adipose tissue (BAT) and beige fat. Each tissue type has distinct metabolic and morphological features (Figure 1). WAT serves as the storage of excess fats and can expand dramatically to accommodate this excess in the form of triacylglycerol [12,13]. The capacity of adipocytes to store excess fat is not unlimited, and chronic increased fat intake and low energy expenditure can cause lipids to accumulate in organs such as the liver, rather than in the adipose tissue [11].

BAT and beige fat have functions distinct from WAT, and are highly metabolically active [14]. BAT and beige fat play important roles in body temperature homeostasis and energy regulation [14]. Adipocytes in BAT have mitochondria that contain uncoupling protein-1 (UCP1). When activated, UCP1 stimulates respiratory chain activity, which increases ATP synthesis to generate heat [13]. The origins and anatomic regions of WAT, BAT and beige fat are also distinct [14]. WAT can be differentiated from mesenchymal cells after the postnatal period in response to excess energy availability. BAT develops embryonically and resides within anatomically defined deposits [13,14]. Interestingly, evidence in mice suggests that the appearance of thermogenic active beige adipocytes in WAT (also referred to as “browning” of white WAT) could be associated with protection against obesity and insulin resistance [13]. These beige adipocytes might originate from the differentiation of precursor cells or conversion of WAT in the presence of PPARγ and PPARα agonists under conditions of increased metabolism and temperature challenges [13]. While all of these adipose types produce factors that regulate metabolic processes, in this review we will focus on factors secreted from adipocytes in WAT and their effects on liver cancer.

## 3. Adipocytes and the Tumor Microenvironment

Healthy adipocytes from WAT store energy and expand while coordinating their activity with other tissues via secretory products. In this regard, adipocytes act as an endocrine organ [15]. A series of morphological, physiological and phenotypic changes can transform a normal adipocyte into a diseased adipocyte known as a cancer-associated adipocyte (CAAS) [15,16]. In contrast to healthy adipocytes, diseased adipocytes are much larger, and expand rapidly but inefficiently with limited vascularity, resulting in a low oxygen environment (hypoxia), massive fibrosis, abnormal remodeling of the extracellular matrix (ECM), alterations in metabolism, and increased inflammation [17]. In response to hypoxia, an angiogenic response is induced, which requires the cooperation between inflammatory adipocytes and stromal cells that promote the production of angiogenic factors VEGF and FGF2, as well as adipokines that enhance a pro-angiogenic microenvironment [18,19]. These inflammatory adipokines include IL-6, TNFα, IL-1β, IL-8, IL-10, monocyte chemoattractant protein (MCP)-1, leptin, adiponectin resistin, plasminogen activator 1, angiotensin II and others [10,20]. This newly developed micro-vascularity further enhances dysregulated growth and expansion. An increase in diseased adipocytes also initiates an increased immune response due to macrophage recruitment and infiltration, resulting in chronic inflammation [21]. Evidence suggests that the observed inflammation is the result of a polarization of macrophages from an M2 to a more pro-inflammatory M1 state, and the subsequent activation of NF-kB and JNK signaling pathways and cytokines [22].

CAASs mostly support a pro-tumorigenic microenvironment through paracrine signaling with the tumor they surround [16]. Studies indicate that obesity-induced hepatocarcinogenesis could be the result of CAAS cells altering distinct molecular mechanisms. This cancer cell–adipocyte crosstalk or paracrine signaling results in structural remodeling, the acquisition of more fibroblast-like phenotypes, and the altered production and release of hormones and adipokines. For instance, histological observations suggest that crosstalk between the epithelial cells of the invading tumor and CAASs may lead to a phenomenon described as the “de-differentiation” of the adipocytes into a fibroblastic-like phenotype [15,23,24]. These fibroblast-type cells are known as cancer-associated fibroblasts (CAFs). They support liver carcinogenesis by providing the framework to remodel the ECM, and, through the secretion of factors that promote angiogenesis and proliferation, decrease immune surveillance and epithelial–mesenchymal transition (EMT) [25]. This promotes cancer cell survival, especially under extreme conditions of stress such as nutrient and oxygen deprivation [22,26]. Moreover, the hypoxic regions in both adipose tissue and tumors activate the hypoxia-inducible transcription factor (HIF) genes and downstream target genes, leading to further proliferation and invasion [27,28]. Since adipose tissue functions as an endocrine organ, the release of these bioactive molecules results in the activation of STAT3 and the PI3K/AKT/mTOR signaling pathway, leading to HCC cell growth [29].

Finally, in advanced stages of cancer, changes in lipid metabolism due to an increased state of inflammation result in adipose tissue reduction and lipolysis known as cancer cachexia [30,31]. The newly released free fatty acids from surrounding adipocytes are taken up and recycled by β-oxidation in the cancer cell mitochondria, and may act to potentiate tumor growth by providing the cancer cells with additional energy substrates in order to survive increased energetic demands [31,32,33]. These findings suggest that adipocytes play an important role in the tumor microenvironment and facilitate HCC development.

Adipocytes can impact the tumor microenvironment and modulate intercellular communication by encapsulating their bioactive materials into extracellular vesicles (EVs). A subtype of EV, the exosomes, has gain significant attention for its contribution to cellular communication and the modulation of diverse biological processes. An important aspect to be discussed is the role of exosomes in the crosstalk between diseased adipocytes and liver cancer cells in order to support a pro-tumorigenic environment. Our primary focus in the next sections will be the role of adipocyte-derived exosomes (AdExos) and HCC-derived exosomes (HCCExos) in the modulation of the tumor microenvironment, as well as the signaling and progression, of HCC.

## 4. Exosomes

EVs are secreted by most cell types and contain proteins, lipids, and different types of RNA such as messenger RNA (mRNA), microRNA (miRNA), long non-coding RNA (lncRNA), circular RNA (circRNA), deoxyribonucleic acid (DNA) and bioactive cell metabolites [34,35]. Exosomes are a subtype of EV, alongside nanoparticles, microparticles, shedding microvesicles, apoptotic blebs, exomers, oncomeres and human endogenous retroviral particles. Each subpopulation of EVs differs according to its size, content, and markers specific to its origin.

In the past three decades, exosomes, which are 30–120 nm small endosomal microvesicles, have been the subject of major consideration [36]. Exosomes can be found in all biological fluids, such as blood [37], urine [38], saliva [39] and breast milk [40]. However, each cell differs in the size and cargo contained in the exosomes they release, which provide prognostic information from the cell of origin and any associated signature markers [41].

The current standards of classification, isolation protocols, detection methods and the overall biology of these vesicles is an active field of investigations, and it is not fully distinguished between EV’s once they reach the extracellular space [42,43]. Even though there are many subclasses of EVs, as we have described, in this review we will be focusing on the literature featuring exosomes, particularly AdExos and HCCExos.

### 4.1. Exosome Biosynthesis

Classically, exosomes are generated by the endocytic pathway. Transferrin receptors at the cell surface bud inward, forming an early stage endosome, and the inward folding leads to the encapsulation of RNAs, cytosolic proteins and bioactive lipids from the cellular environment, as well as the maturation of the early endosome into the late endosome [44]. Characteristically, late endosomes, also known as microvesicular bodies (MVBs), engage the action of the Endosomal Sorting Complex Required for Transport (ESCRT), which is one of the main signaling networks responsible for the sorting of cargo within MVBs and membrane deformation. Within the MVB, pinching and deformation of the limiting membrane leads to the creation of intraluminal vesicles (ILVs), which are exosome precursors in the MVB lumen (Figure 2) [45]. The ESCRT family is comprised of approximately 30 proteins, assembled in four complexes. These complexes are designated ESCRT 0, I, II and III. The ESCRT complexes are recruited to the cytosolic side of the endosomal membrane for the sorting of selected proteins, a process which requires the ubiquitination of the cytosolic tail of their receptors [46]. Transmembrane proteins in the endosomal membrane are first captured by ESCRT 0, then the ESCRT I complex binds to the ubiquitinated cargo proteins and activates the ESCRT II complex [45]. Thus, ESCRT I and II are responsible for the inward bud formation with sorted cargo. ESCRT II, in turn, initiates the oligomerization and formation of the ESCRT III complex (Figure 2). Exosomal proteins such as Alix are associated with ESCRT through the tumor suppressor susceptibility gene 101 (TSG101), charged multivesicular protein body 4 (CHMP4), and other proteins that are involved in the budding process and exosomal cargo selection [47]. Finally, ESCRT III forms a constriction at the budding neck. Complete scission at the budding neck is facilitated by the vacuolar protein sorting 4 (VPS-4) complex, leading to the formation of ESCRT-III-catalyzed membrane fission [46].

However, studies have demonstrated the formation of ILVs in the absence of the ESCRT complexes, which indicates the presence of ESCRT-independent mechanisms. Tetraspanin-dependent exosome formation, involving tetraspanins such as CD9, CD63 and CD82, and lipid-dependent exosome formation and release have been investigated in addition to the action of the ESCRT complexes. For example, the sorting of cargo has been linked to CD82-rich domains, whereas CD9 depletion results in a decrease in exosome release [48]. Moreover, lipids have been shown to be contributors of cargo sorting into ILVs. Cholesterol, for instance, is found in quantities within the MVB lumen, and has been implicated in exosomal content sorting. The ceramide-dependent exosome biogenesis pathway has also been found to be significant. Ceramide is synthesized by the catalytic action of neutral sphingomyelinases (nSMases) from sphingomyelin [49]. Generally, when GW4869, a nSMase inhibitor, was used to treat Oli-neu cells, exosome release was significantly reduced [50], thus confirming that ceramide is actively involved in exosome biogenesis. Therefore, lipid increase is a driver of exosome biogenesis and release. Interestingly, leptin, an important obesity-related protein, as described before, has been demonstrated to be a driver of exosome biogenesis in a recent breast cancer study by Giordano et al. [51]. These findings reinforce the link between lipids and tumor microenvironment deregulation through the release of exosomes.

Due to their endocytic origin, both exosomes formed by ESCRT-dependent and –independent pathways are often enriched in endosome-associated proteins such as Rab proteins, SNAREs, Annexins and Flotillin [52]. Rab proteins in particular play a critical role in intracellular vesicular transport [53]. Rab proteins belong to a family of about 60 small GTPases. Each of the Rab GTPases, such as Rab5a, Rab5b, Rab27a and Rab27b, is preferentially associated with an intracellular compartment involved in vesicle budding and mobility through interaction with the cytoskeleton or tethering with the membrane of an acceptor compartment [53]. Upon complete maturation in MVBs, exosomes are released into the tumor microenvironment (Figure 2).

### 4.2. Exosome Secretion and Cell Signaling

Exosomes are secreted from most cell subtypes, such as inflammatory cells, stem cells, muscle, neurons, epithelial cells and adipocytes [54,55]. Exosomes are stable in the extracellular environment [54], and can mediate intercellular communication by transferring and unloading a variety of macromolecules in the target cells, resulting in a wide range of cellular responses, including changes in gene expression [55,56].

Most exosomes resist membrane breakdown and, therefore are ideal carriers for short RNAs. They can persist in the extracellular microenvironment for extended periods [54]. In contrast, certain exosomes have a membrane that quickly recognizes and fuses with their target cell membrane to release their bioactive cargo [57]. Studies of exosome transport demonstrate that exosomes can target specific cells within a tissue [54]. Some exosomes have been shown to distinguish receptors on the surface of the target cells for selective binding [54].

Once in proximity to their target cells, exosomes undergo fusion with the endocytic or plasma membrane. Subsequently, vesicle membranes integrate with fusion membranes and discharge their exosomal cargo into the cytosol [54]. For example, exosomes derived from fibroblasts (which are adipocyte precursors) are first recruited on filopodia, but then navigate to the cell surface until they reach specific endocytic sections of the cell membrane dedicated to internalization [58]. Thus, fusion with the plasma membrane occurs at these endocytic hotspots. Exosomal cargo molecules can be discharged in the cytosolic layer near the surface of the cell or, after docking, exosomes can be transferred into the intracellular compartment, where they participate in intracellular trafficking. Both physiological and pathological conditions are attributed to the effect of exosomal molecules in target cells.

## 5. Adipocyte-Derived Exosomes (AdExos)

The vast majority of exosomes from adipose tissue are adipocyte-derived (AdExos) [59]. Recent evidence reveals that AdExos contain biomolecules such as miRNAs, lncRNAs, lipids, proteins, etc., which are vital to intercellular organ crosstalk through the regulation of various pathways [60,61,62,63]. Here, we give a thorough introduction to how these components exert their functions in liver diseases, including chronic inflammation, fatty liver diseases and HCC.

### 5.1. AdExos miRNAs and Chronic Inflammation

Obesity is characterized by chronic adipose tissue inflammation. Inflammation promotes systemic tumor growth factor β1(TGF-β1) levels, resulting in increased myofibroblasts, which are implicated in increased extracellular matrix proteins. The accumulation of myofibroblasts and extracellular matrix proteins contributes to fibrotic disease, such as NAFLD. Importantly, Wnt/β signaling is necessary for TGF-β1-mediated fibrosis. Ferrante et al. showed that the four AdExos miRNAs, miRNA-23b, miRNA-148b, miRNA-4269 and miRNA-4429, are capable of regulating TGF-β1 and Wnt/β signaling by downregulating activin receptor type-2B (ACVR2B) (Table 1) [64]. Their findings follow a previous study that showed that exosomes from visceral adipose tissue induce dysregulation of the TGF-β pathway in the pathogenesis of NAFLD [65]. On the other hand, inflammation is related to lymphangiogenesis, which is a key factor of inflammatory pathophysiological processes, like inflammatory bowel disease (IBD). Lymphangiogenesis could facilitate the stabilization of inflammatory cells and resolve tissue edema in acute and chronic inflammatory situations. Similarly, studies also showed that exosomes isolated from adipose-derived stem cells treated with vascular endothelial growth factor-C (VEGF-C) expressed more miRNA-132, which promotes lymphangiogenesis by regulating TGF-β/Smad signaling [66].

Immune cell infiltration is a crucial hallmark of inflammation, and includes a variety of cell types such as macrophages, T cells, B cells, neutrophils and eosinophils [67]. Macrophages exhibit two distinguished phenotypes: classically activated, pro-inflammatory M1, and natively activated, anti-inflammatory M2 [68]. The transformation of macrophages has been recognized as a vital factor in the formation of adipose tissue and the development of chronic inflammation. Studies have demonstrated that miRNAs within exosomes, such as miRNA-34a, promote macrophage activation via different biological pathways [68]. Zhang et al. demonstrated that M1 macrophages were activated by AdExos miRNA-155, thus resulting in chronic inflammation and local insulin resistance [69]. A similar conclusion was drawn by in vivo experiments, and the study revealed that miRNA-155 knockout mice displayed an attenuated obesity-induced glucose intolerance and systemic insulin resistance [70] (Table 1).

### 5.2. AdExos Carriers and Lipolysis

Other than miRNAs, AdExos carry glycosylphosphatidylinositol-anchored (c)AMP-degrading phosphodiesterase (Gce1), and are later phagocytosed by smaller adipocytes, during which free fatty acids are made available to accelerate the synthesis of triglycerides [71]. This finding revealed independent canonical lipolysis of lipid release and delivery by AdExos [71]. More than an exciting novel mechanism of lipolysis, this report also confirmed that AdExos cause lipid accumulation in macrophages and the differentiation of bone marrow progenitors into adipose tissue macrophage-like cells, which results in lysosomal biogenesis and a transcriptional program specific to adipose tissue macrophages [59].

### 5.3. AdExos Carriers and Tumorigenesis

The tumor microenvironment is essential for tumor occurrence and progression. Adipocytes, as the main cellular components of adipose tissue, are responsible for energy homeostasis and have been linked with tumorigenesis [88,89]. Specifically, studies have shown that AdExos contribute to tumor invasion via their influence on the intercellular space. AdExos derived from 3T3-L1 cells contain high levels of MMP3. In turn, the activation of MMP9 by MMP3 promotes invasion in vitro and in vivo [72]. These proteins are involved in lipid metabolism, and also modulate melanoma migration through metabolic reprogramming and increasing mitochondrial number and density [72,73]. A recent study demonstrated that exosomes originating from adipose-derived mesenchymal stem cells (MSCs-AdExos) contain a total of 1185 protein groups [74]. Gene ontology analysis revealed that in addition to protein binding, these protein groups are involved in various mechanisms, such as posttranslational modification, protein turnover, chaperone function in metabolic pathways, focal adhesion, regulation of the actin cytoskeleton, microbial metabolism, and tissue repair-related signaling pathways [74]. Indeed, in addition to miRNAs, lipids and proteins contained in AdExos are also implicated in the progression of numerous pathologies. These studies demonstrate the exosome-driven mechanisms by which adipocytes could promote tumorigenesis by packaging the oncogenic molecules uptaken by the receiving cells.

## 6. AdExos Mediated Cell Signaling Progression in HCC

Exosomes are considered signaling bodies that mediate core biological processes, such as immune surveillance, the suppression of inflammatory responses, the maintenance and plasticity of stem cells, and the proliferation and inhibition of apoptosis between neighboring and distal cells [90,91,92]. For example, immune cells utilize exosomes as antigen-presenting vesicles, stimulate anti-tumoral immune responses and suppress inflammation [91]. Thus, exosomes can foster cancer progression by modulating communication between diseased and healthy cells through their microenvironment, promoting adverse immune tolerance and supporting mechanisms of survival and resistance [90].

Adipocytes provide metabolic substrates to tumors and appear to be significant in metastasis and tumor progression. Studies on exosomes and cancer are recent, thus some conflicting data needs to be resolved. Numerous studies suggest that MSCs-AdExos or exosomes released from pre-adipocytes regulate cancer tumor cell behavior [75,76] (Table 1). When AdExos from 3T3-L1 pre-adipocytes are injected into MCF10-DCIS, breast cancer cell tumor growth is promoted (in vivo), whereas the exposure of MCF-7 breast cancer cells to MSCs-AdExos increased cell migration and tumor progression by regulating the Wnt/β-catenin pathway [75,76]. On the other hand, in SKOV-3 and A2780 ovarian cancer cells, MSCs-AdExos inhibited cell proliferation, colony-forming abilities, wound-repair activities and cell survival [77]. The study further demonstrated that MSCs-AdExos contain numerous miRNAs that induce the expression of Bax, caspase 3, caspase 9 and apoptosis markers, and downregulate the expression of anti-apoptotic molecule Bcl-2, leading to the inhibition of cell proliferation and cell survival in ovarian cancer cells [77].

However, the role of the AdExos-mediated regulation of NAFLD and HCC cell signaling is poorly studied so far. It has been shown that AdExos circulating miRNAs regulate gene expression in specific distant organs, leading to increased intercellular communication [93]. An earlier study revealed that AdExos deregulate the TGF-β pathway and induce fibrotic signaling in hepatic cells in models of obesity [65]. Interestingly, exosomes derived from pancreatic ductal adenocarcinomas trigger liver pre-metastatic niche formation in naive mice [62], and a pre-metastatic niche paves the way for a higher liver metastatic burden [62]. Thus, it is not clear whether AdExos stimulate pre-metastatic niche formation in the human liver in conditions of obesity. For example, miRNA-23a/b levels were increased in mature adipocytes compared to pre-adipocytes and exosomes mediated the transfer of this miRNA from adipocytes to HCC cells, resulting in a more aggressive phenotype by targeting the tumor suppressor Von Hippel–Lindau (VHL) [78].

A new type of noncoding RNAs, circRNAs, has also been investigated due to its ability to act as a “sponge” and bind miRNAs regulating gene expression [79]. Higher levels of circRNA, which regulates deubiquitination (circ-BD) and ubiquitin-specific protease 7 (USP7), were found in exosomes isolated from patients with elevated body fat ratios (Table 1) [79]. Higher circ-BD was encapsulated in exosomes isolated from adipocytes. Furthermore, the treatment of HCC cells with exosomes harvested from mature adipocytes resulted in a decrease of miRNA-34a (tumor suppressor), an increase in the USP7/Cyclin A2 signaling pathway (pro-oncogenic), increased HCC cell proliferation, and reduced DNA damage in HepG2 cells. However, the knockdown of circ-BD abrogated these effects [79]. Exosomes isolated from the blood of mice injected with HCC cells with stable circ-BD overexpression indicated a pronounced increase of this circRNA and increased tumor growth in comparison to mice injected with knockdown cells [79]. In a separate study, Liu et al. isolated exosomes from HCC patients with high body fat and discovered that these had higher levels of miRNA-231/b when compared to exosomes from patients with lower body fat [78]. These exosomes were effectively taken up by HCC cells, leading to increases in cell migration and chemoresistance. Moreover, miRNA-23a/b effectively targeted and decreased levels of VHL (von Hippel–Lindau) tumor suppressor protein, affecting its downstream targets including HIF1α (Table 1) [78]. In vivo studies showed that exosomes isolated from mice on a high-fat diet had increased miRNA-23a/b levels, and also showed increased tumor growth, increased levels of VEGF, GLUT1, and HIF1α, and decreased levels of VHL. Mice implanted with cells with stable knockdown of this miRNA showed increased tumor growth and an abrogation of effects previously observed in obese mice [78].

Finally, adipocytes also produce adiponectin, a pleiotropic organic-protective protein, which has been found to enhance AdExo biogenesis and secretion. Thus, adipose tissue metabolism regulates exosome production throughout the body [94]. Mori et al. [95] and Obata et al. [96] showed that endosomal vesicles of aortic endothelial cells contained adiponectin. They suggested that adiponectin binds to T-cadherin both in vitro and in vivo, forming an adiponectin/T-cadherin system responsible for regulating AdExo biogenesis. This complex stimulates AdExo biogenesis and the release of ceramides from exosomes, which reduces ceramides within the cell [96]. As aforementioned, ceramide is a lipid essential for the formation of exosomes through an ESCRT-independent mechanism, thus the role of adiponectin in regulating its efflux is crucial [96].

## 7. HCCExos Mediated Cell Signaling and Progression in HCC

HCC cells could further promote oncogenesis and metastasis by using exosomes to alter their microenvironment. Exosomes secreted by cancer cells act as autocrine and paracrine communicators to modulate extracellular communication and educate neighboring cells to promote angiogenesis, EMT and chemoresistance [97]. Studies have demonstrated that HCC cell interaction with surrounding cells in the microenvironment through exosomes plays a crucial role in the development of a more aggressive phenotype [80,98]. Exosomes could also be utilized as potential prognostic markers for HCC, depending on their cargo [99]. Exosomes isolated from HCC cells are effectively taken by adipocytes, altering their inflammatory phenotype. Interestingly, adipocytes treated with HCC exosomes increased migration and tube formation in vitro, promoted angiogenesis and tumor growth, and increased macrophage recruitment in a mouse xenograft model [80]. HCCExos can also increase tumorigenicity in normal cells by modulating cell signaling in their microenvironment (Table 1 and Figure 3). HCCExos derived from the HCC cell line HepG2, when exposed with adjacent adipocytes, are actively internalized into adipocytes and induce inflammatory cytokine secretion, the activation of Nf-kB and other kinases, and induce adipocyte signaling to support HCC tumor growth and progression [80]. HCCExos contain unique miRNAs that are delivered to recipient cells through HCCExos, thus activating transforming growth factor β activated kinase-1(TAK1) expression, and modulating nuclear factor (NF)-κB and c-Jun NH2-terminal kinase (JNK)/p38 MAPK networks in target cells [81]. The study further suggests that HCCExos-mediated miRNAs transfer can regulate intercellular communication in HCC cells, leading to an increase in HCC tumorigenesis and local spread in the liver [81]. A similar study proved that HCCExos contain several pro-tumorigenic RNAs and proteins, such as MET proto-oncogene, the caveolins, and S100 family members. When these HCCExos are added to the normal hepatocyte, they activate PI3K/AKT and MAPK signaling pathways and increase secretion of MMP-2/MMP-9 in host hepatocyte cells, which increases tumorigenesis in normal hepatocytes [82].

HCCExos not only modulate tumorigenesis, but also promote angiogenesis. HCCExos derived from cancer stem cell-like CD90+ liver cells contain lncRNA H19. Human umbilical vein endothelial cells (HUVECs) exposed to HCCExos, containing lncRNA H19, promote angiogenic phenotype and cell-to-cell adhesion by increasing the transcription of VEGF, a potent proangiogenic cytokine [83]. Moreover, tumor-derived exosomes carry bio-functional molecules that activate mesenchymal-associated gene expression and induce EMT signaling in host cells, as well as increased transformation, invasion/migration, angiogenesis, and pre-metastatic niche formation [100,101]. When exposed to low metastatic HLE cells, HCCExos isolated from metastasis-prone MHCC97-H cells augmented the expression of mesenchymal markers such as N-cadherin, α-SMA and vimentin, and decreased expression of E-cadherin, an epithelial marker. Activated MAPK/ERK signaling was also observed, indicating that HCCExos upregulates EMT transitions in HLE cells [84]. A recent study indicates that HCCExos regulate CAFs by delivering miRNA-1247-3p to normal fibroblasts. In normal fibroblasts, HCCExos miRNA-1247-3p targets β-1,4-galactosyltransferases (B4GALT3), activates β1-integrin-NF-κB signaling, and converts normal fibroblasts into CAFs [85]. The study further suggests that activated CAFs proceed in secreting pro-inflammatory cytokines, such as IL-6 and IL-8, enhancing HCC progression and metastasis [85]. Similar studies further confirm the effectiveness of short RNAs as HCCExos cargo [86,87]. It was shown, for example, that HCCExos derived from HCC Huh7 cells carry miRNA-122. The inhibition of cell growth and cell proliferation was observed in recipient HepG2 cells when these miRNA-122 enriched HCCExos were exposed to miRNA-122-deficient HCC HepG2 cells [86]. On the other hand, HCCExos-derived ultraconserved lncRNA TUC339, when exposed to HCC cell lines HepG2, Hep3B and PLC/PRF5 within the microenvironment, enhanced HCC cell proliferation and migration [87]. Indeed, this evidence suggests that AdExos and HCCExos carry miRNAs, lncRNA and several key proteins, and when these biomolecules are delivered to HCC recipient cells through exosomes, they activate a cell signaling cascade in neighboring HCC cells and modulate tumorigenesis.

## 8. Exosomes as Therapeutic Vehicles

Exosomes are also being explored as therapeutic vehicles for the treatment of HCC [102]. Since MSCs can produce a large number of exosomes, MSCs could potentially be utilized as therapeutic vehicles [103]. For example, miRNA-122 has been demonstrated to promote chemosensitivity in HCC, and treatment with miRNA-122-enriched exosomes altered gene expression in recipient HCC and increased chemotherapeutic drug sorafenib sensitivity both in vitro and in vivo [103]. Another group utilized MSCs-AdExos to package miRNA-199a-3p, which has been shown to reduce HCC cell aggressiveness and enhance chemosensitivity to doxorubicin [104]. Furthermore, treatment with these exosomes altered the mTOR pathway and decreased chemoresistance in these cells. They observed similar effects in-vivo, confirming the utility of engaging exosomes as therapeutic vehicles [104]. A separate study utilizing a rat model of HCC also demonstrated the benefit of exploiting MSCs-AdExos as therapeutic vehicles. In this case, they employed exosomes isolated from healthy animals to treat diseased rats. They observed a marked reduction in disease progression, as evidenced by a decrease in tumor volume and an increased immune response [105]. Furthermore, MSCs-AdExos activate natural killer T-cell (NKT-cell) antitumor responses, increase early apparent diffusion coefficient (ADC), and low-grade tumor differentiation leads to a reduction in HCC tumor growth in rats [105]. Similarly, the exposure of mouse hepatic stellate (HST-T6) cells to miRNA-181-5p-modified MSCs-AdExos downregulated collagen I, vimentin, a-SMA and fibronectin in the liver, and prevented liver injury and fibrosis [106]. The study further pointed out that overexpression of miRNA-181-5p MSCs-AdExos downregulates Stat3 and Bcl-2 expression, as well as the induction of autophagy in the HST-T6 cells [106]. These studies evidence the importance of understanding how exosomes change the dynamics of the microenvironment and their possible utility to effectively treat this disease.

## 9. Conclusions

In this review, we analyzed the association between obesity and adipose-derived factors and their role in HCC progression. We also summarized the recent literature regarding adipocyte processes such as adipogenesis, exosome biosynthesis, and the encapsulation of a variety of molecules in exosomes and their release into recipient cells—processes that play key roles in tumorigenesis. Studies demonstrate that AdExos carry key miRNAs and lncRNAs, and upon their delivery into recipient HCC cells, modulate HCC host cell signaling, thus changing the microenvironment. Furthermore, HCCExos regulate signaling with neighboring cells and organs, such as in adipose tissue and liver. Several key issues remain unanswered, such as how miRNA, lncRNA and proteins are sorted into exosomes, and how the uptake process for cell-specific exosomes is regulated.

Body fluids of cancer patients, like plasma, serum and urine, are excellent sources of exosomes for research [107]. However, isolation and purification to obtain a high yield of exosome material are challenging and expensive, which can limit exosome research. Thus, the identification of markers that are both sensitive and specific for HCC is imperative in order to enable the use of these microvesicles as therapeutics. Studies have also revealed that AdExos and HCCExos also have the potential to be diagnostic biomarkers for early-stage HCC, and that HCCExos miRNAs, lncRNAs and proteins could be established as new biomarkers for monitoring HCC development and progression [99,107,108]. Finally, since AdExos can result in HCC suppression, the characterization of AdExos cargo will potentially provide novel approaches to the treatment of HCC through exosome delivery [109]. To exploit exosomes as carriers of biological and pharmacological cargo, new therapeutic agents that target HCC and can be easily encapsulated into exosomes should be explored.

## Figures and Tables

**Figure 1 cells-09-01988-f001:**
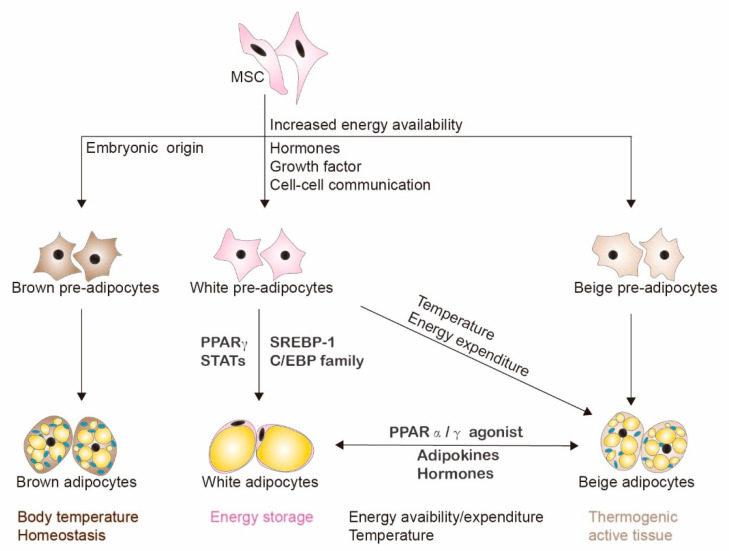
Adipogenesis and different types of adipose tissue. Adipocytes can be divided into three different types depending on their origin, metabolic activity and morphological features. These are brown (BAT), white (WAT) and beige adipocytes. Mesenchymal precursors are committed and differentiate into pre-adipocytes, then further mature into adipocytes of a particular lineage influenced by various transcription factors, cell to cell communication, and extracellular signaling. WATs can also be transformed into beige adipocytes and vice versa, influenced by energy availability, temperature and extracellular signaling.

**Figure 2 cells-09-01988-f002:**
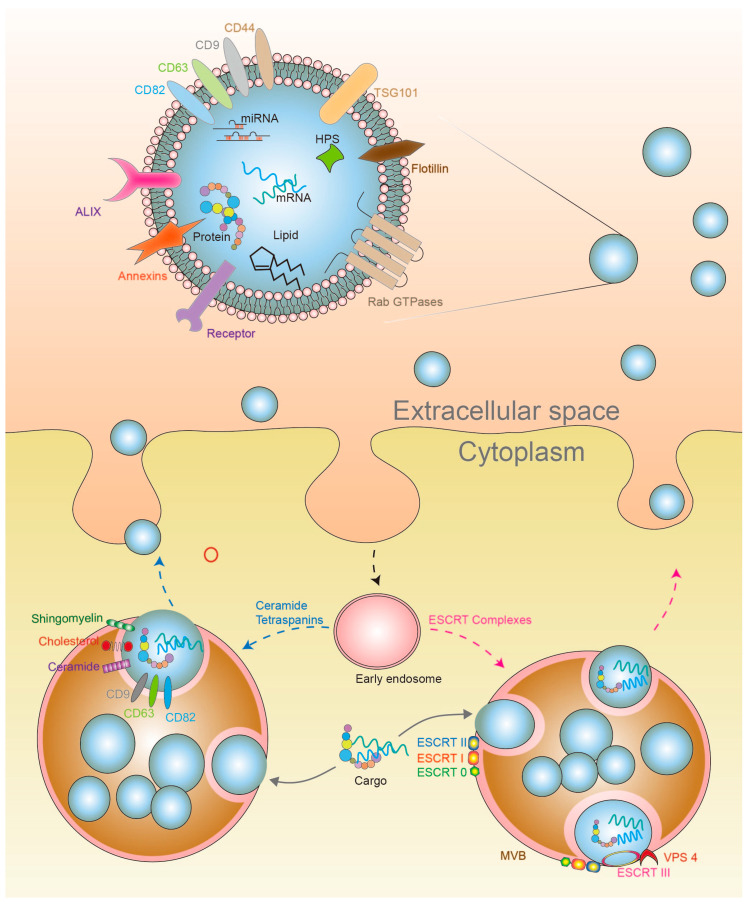
Exosome biogenesis and release. Exosomes are generated by the endocytic pathway activated by local signaling that culminates in its exocytosis into the extracellular microenvironment. Inward budding and complete invagination results in the formation of MVBs and the encapsulation of RNAs, cytosolic proteins, metabolites, hormones and bioactive lipids. After further processing, a fusion of the membrane facilitates the release of mature exosomes into extracellular space. Alternatively, they may also form lysosomes that are slated for degradation.

**Figure 3 cells-09-01988-f003:**
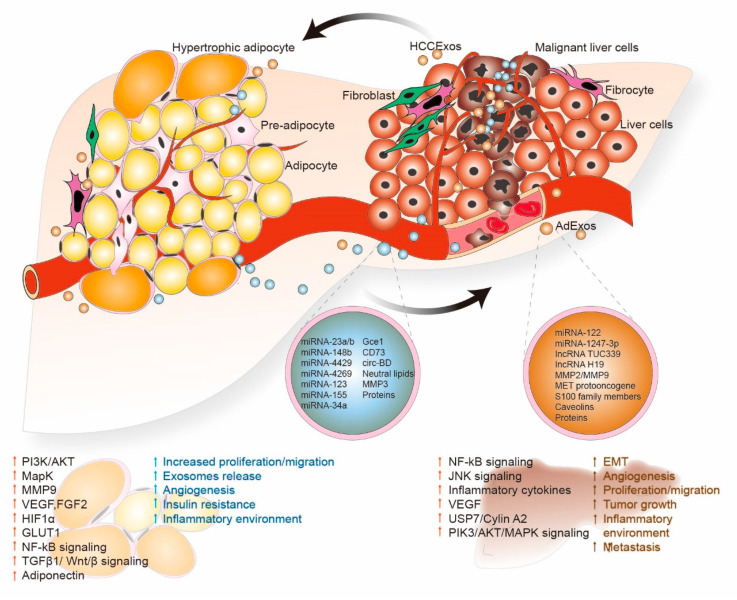
Adipocyte and HCC exosomes crosstalk leading to increased tumor progression. Exosomes released by both adipocytes (AdExos) and HCC (HCCExos) have unique intracellular components, including a wide variety of mRNAs, microRNAs, lncRNAs, circRNAs, lipids, metabolites and proteins. These intracellular components are utilized as autocrine or paracrine communicators to induce pathways that result in increased cell proliferation, angiogenesis, invasion, and other hallmarks of malignancy.

**Table 1 cells-09-01988-t001:** Exosome source, cargo and function in disease.

Exosome Source	Exosomal Cargo	Cell Signaling and Biological Function in Human Disease	Ref No.
**AdExos**	miRNA-23b, miRNA-148b, miRNA-4269, miRNA-4429	AdExos from obese patients downregulates ACVR2B and regulates TGF-β1/Wnt/β signaling in A549 cells.	[64]
**AdExos**	miRNA-148b, and miRNA-4269, miRNA-23b and miRNA-4429	AdExos from visceral adipocyte tissue dysregulates TGF-β family members in hepatic stellate cells and HCC HepG2 cells.	[65]
**AdExos**	miRNA-132	EdExos from VEGFC/adipocyte derived stem cells increased proliferation/migration/tube formation and lymphangiogenesis by targeting Smad-7 and regulating TGF-β/Smad signaling.	[66]
**AdExos**	miRNA-34a	Incubation of AdExos with bone marrow-derived macrophages inhibited IL-4-induced M2 macrophage polarization by directly targeting Krüppel-like factor 4 (Klf4).	[68]
**AdExos**	miRNA-155	HFD increased accumulation of AdExos miRNA-155 and polarization of M1 macrophages led to increased inflammation and insulin resistance in obese mice.	[69,70]
**AdExos**	Gce1, 5′-nuceotidase CD73	Gce1 and CD73 are released from adiposomes to intracellular lipid droplets of the acceptor adipocytes upregulates esterification of fatty acids into triacylglycerol.	[71]
**AdExos**	Neutral lipids	Obese mice released more exosomes than lean mice. These exosomes provided precursor lipids such as acylglyceride, inducing the differentiation of bone marrow progenitors into adipose tissue macrophages-like cells.	[59]
**AdExos**	MMP3	AdExos from 3T3-L1 increased MMP9 activity and metastasis in 3LL lung cancer cells.	[72]
**AdExos**	Variety of proteins	Exposure of AdExos isolated from obese individuals or from 3T3-F442A cells regulates migration/invasion through metabolic programming in SKMEL28 and 1205Lu melanomas.	[73]
**MSCs-AdExos**	1185 protein groups	MSCs-AdExos cargo can regulate metabolism, motility, tissue repair, protein turnover, chaperoning and post transcriptional modifications.	[74]
**AdExos (pre-adipocytes)**	miRNA-140, SOX9 and other oncogenic growth factors and cytokines	AdExos increased migration/proliferation/mamosphere formation and breast cancer tumor growth in vivo.	[75]
**MSCs-AdExos**	Not specified	MSCs-AdExos increased proliferation/migration by activation of the Wnt signaling pathway in MCF7 breast cancer cells.	[76]
**MSCs-AdExos**	miRNA-4792, miRNA-320b, miRNA-320a and other miRNAs	MSCs-AdExos decreased proliferation/wound-repair/colony formation by increasing apoptosis in ovarian cancer cells.	[77]
**AdExos**	miRNA-23a/b	Treatment of various HCC cell lines with ADExos promoted tumor growth in vivo by downregulation of VHL.	[78]
**AdExos**	circ-BD (circRNA)	Higher levels of circ-DB in AdExos from obese HCC patients correlate with a decreased in miRNA-34a, activation of USP7/Cyclin A2 signaling led to increased HCC aggressiveness.	[79]
**HCCExos**	Various proteins	HCCExos internalized by adipocytes resulting in increased inflammatory cytokine secretion, NF-κB signaling, proliferation/migration, and tumor growth in vivo.	[80]
**HCCExos**	miRNAs	HCCExos promotes inter-cellular communication and aggressiveness by TAK1 expression and by modulation of JNK/p38 MAPK and NF-κB signaling pathways.	[81]
**HCCExos**	MET protooncogene, S100 family members and caveolins	Exposure of HCCExos to non-motile MIHA cells activated PI3K/AKT/MAPK signaling, increased secretion of MMP-2 and MMP-9, and lead to increased migration/invasion and motility.	[82]
**HCCExos**	lncRNA H19	HCCExos from Huh7-CD90+ cells promoted transcription of VEGF, angiogenesis and cell adhesion in HUVECs.	[83]
**HCCExos**	Pro-tumorigenic RNAs and proteins	HCCExos from highly metastatic HCC MHCC97H cells increased migration/chemotaxis and EMT through the MAPK/ERK signaling in low metastatic HCC cells	[84]
**HCCExos**	miRNA-1247-3p	HCCExos from highly metastatic HCC cells promoted the conversion of fibroblasts into cancer-associated fibroblasts and secretion of inflammatory cytokine by targeting B4GALT3 and activating β1-integrin-NF-κB signaling.	[85]
**HCCExos**	miRNA-122	HCCExos from Huh7 reduced cell proliferation and cell growth in HepG2 HCC cells.	[86]
**HCCExos**	lncRNA TUC339	HCCExos promoted HCC cell growth and adhesion by modulating local tumor environment.	[87]

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
