# Peer review of "The Role of Exosomes in the Crosstalk between Adipocytes and Liver Cancer Cells"

_cells, 2020, doi:10.3390/cells9091988_

Round 1
Reviewer 1 Report
The review entitled « The Role of Exosomes in the Crosstalk Between 2 Adipocytes and Liver Cancer Cells” and submitted by Rios-Colon at al. is organised in logical order and is very didactic. It summarises the genesis of adipocytes, the role of adipocyte in generating tumour microenvironement, the genesis of exosomes, the content of adipocyte derived exosomes and HCCexosomes and their role in the progression of HCC.
I will have few suggestions to improve the review.
- Paragraph 4.1 Exosomes biosynthesis:
The ESCRT pathway is well described, however the generation of exosomes through non-ESCRT dependant pathway should also be briefly mentioned in this paragraph.
To present in order of events, the paragraph describing the MVBs starting line 161 finishing line 176 could be separated in 2: (1) The generation of late endosomes and formation of MVBs should be introduced before the description of ESCRT pathway (line 161-165). (2) The fusion of the MVBs with the membrane should be at the end of paragraph 3 as it is currently (line 165-176).
Figure 2: Figure is not fully accurate as ESCRT 0-II complexes seem to be floating in the cytosol of the cells, yet these complexes are associated with membrane.
Figure 3: This figure 3 could be a key figure for the review. Thus it should reflect and summarize well paragraphs 5 and 6. In this context, few elements described in the text are missing in this figure: for example the authors described well in the role of AdExos miRNAs, miRNA-23b, miRNA-148b, miRNA-4269, and miRNA-4429 218 in the regulation of regulation of TGF-β1/Wnt/β pathway, but this element is missing in figure 3.
In the text it is clearly explain that exosomes extracted from high fat diet mice have an impact on tumour growth and increase the expression level of VEGF; GLUT1 and HIF1a, but in figure 3 neither tumour growth nor VEGF are mentioned while GLUT1 and HIF1a are.
Leptin is mentioned in the text in paragraph “3. Adipocytes and the tumor microenvironment” and figure 3, while the other inflammatory adipokines are mentioned in the text only. Yet Leptin is not mentioned in the text to be secreted through exosomes nor to be activated by exosomes.
The effect of adiponectin to stimulate the biogenesis of exosomes by adipocyte is also missing in the figure.
The stimulation of the inflammation is not mentioned in the figure, while it is well described in the text for both AdExo and HCCexo. Idem for cell proliferation and spreading.
Generally, the pathways affected by AdExo and HCCexo and mentioned in Figure 3 are a mixed of specific pathway (eg: MMP2-9, or PI3K/AKT) or more generic/global cellular event (eg: angiogenesis, EMT). It may help if the authors could separate the 2 levels and show the impact of specific pathways on generic/global cellular event when possible.
The zoom in AdExo show that these exosomes contain RNA, DNA, Metabolites, lipids etc, yet in the text and Table 1 all these elements are not mentioned (eg DNA or metabolites). Either the authors expand on these molecules missing in the text and their role in AdExo or HCC exosomes in the context of HCC, either the list of elements present in these exosomes should be reduced. Could the molecules frequently observed across studies cited in the text and Table 1 be named in figure 3 (eg miRNA148b, miRNA4269, miRNA23b, miRNA4269, etc)? Would it be possible to specify the content of HCCexo as well in the figure? This would highlight the specificity of each exosome subpopulation.
It would be good if the authors could find a way to highlight the impact on different pathways on adipocytes by the HCCexosomes (I’m guessing it is the list on left side of the graph), and the impact of AdExo on liver cells (I’m guessing it is the list on the right side): either using a colour coding or moving the list for HCCexosomes on the top left of the schema as the HCCexo are on the top of the schema.
I suggest to the author to revise this figure and make it as complete as possible to reflect their text, as this figure could be a key one for their review that would stick in the mind of any readers and could be cited and mentioned later in conferences etc.
Minor: In the introduction define HCC again.
Line 151: typo: "ESCORT I" should be "ESCRT I"
Line 213: “AdExos miRNAs and chronic inflammation:” it seems this is a new subparagraph, but probably not correctly formatted.
Author Response
August 20, 2020
Re: The Role of Exosomes in the Crosstalk between Adipocytes and Liver Cancer Cells
Reviewer 1
The review entitled « The Role of Exosomes in the Crosstalk Between 2 Adipocytes and Liver Cancer Cells” and submitted by Rios-Colon at al. is organised in logical order and is very didactic. It summarises the genesis of adipocytes, the role of adipocyte in generating tumour microenvironement, the genesis of exosomes, the content of adipocyte derived exosomes and HCCexosomes and their role in the progression of HCC.
We want to thank reviewer 1 for emphasizing the content and contribution of our manuscript to the field.
I will have few suggestions to improve the review.
- Paragraph 4.1 Exosomes biosynthesis:
The ESCRT pathway is well described, however the generation of exosomes through non-ESCRT dependant pathway should also be briefly mentioned in this paragraph.
We agree with the reviewer. To further clarify exosome biogenesis, we have included a paragraph briefly mentioning different types of ESCRT-independent exosome biogenesis mechanism which can be found in the revises section “4.1. Exosome Biosynthesis”, lines 141-190.
- To present in order of events, the paragraph describing the MVBs starting line 161 finishing line 176 could be separated in 2: (1) The generation of late endosomes and formation of MVBs should be introduced before the description of ESCRT pathway (line 161-165). (2) The fusion of the MVBs with the membrane should be at the end of paragraph 3 as it is currently (line 165-176).
Changes have been made to better reflect the process of exosome biogenesis from endocytosis, to early and late endosome formation/MVB formation, to ILV formation and cargo sorting, to exosome release. Thus, section “4.1 Exosome Biogenesis” has been modified.
- Figure 2: Figure is not fully accurate as ESCRT 0-II complexes seem to be floating in the cytosol of the cells, yet these complexes are associated with membrane.
We have modified Figure 2 to accurately represent the ESCRT complexes associated with the membrane. We have further modified this figure to also show the ESCRT-dependent and -independent pathways.
- Figure 3: This figure 3 could be a key figure for the review. Thus it should reflect and summarize well paragraphs 5 and 6. In this context, few elements described in the text are missing in this figure: for example the authors described well in the role of AdExos miRNAs, miRNA-23b, miRNA-148b, miRNA-4269, and miRNA-4429 218 in the regulation of regulation of TGF-β1/Wnt/β pathway, but this element is missing in figure 3.
We recognize that figure 3 is a key figure of this publication and thus we are incorporating these suggestions to our figure. We have modified figure 3 to accurately reflect paragraphs 5 and 6. Changes made included more accurate exosomal content as pointed below. We have also included the downstream targets or more systemic effects in a separate, color coded, section of figure 3.
In the text it is clearly explain that exosomes extracted from high fat diet mice have an impact on tumour growth and increase the expression level of VEGF; GLUT1 and HIF1a, but in figure 3 neither tumour growth nor VEGF are mentioned while GLUT1 and HIF1a are.
We have included both the contribution of VEGF and also the effects of tumor growth in the modified Figure 3.
Leptin is mentioned in the text in paragraph “3. Adipocytes and the tumor microenvironment”and figure 3, while the other inflammatory adipokines are mentioned in the text only. Yet Leptin is not mentioned in the text to be secreted through exosomes nor to be activated by exosomes.
The significance of adipokines, including leptin, in the development of disease and their effects in multiple organs have been documented including liver cells [Chen et al, 2017; DOI: 10.1016/j.yexcr.2017.04.022O]. There is evidence that leptin increases exosome number in conditioned media of two different breast cancer cellular models through the upregulation of Tsg101 [Giordano et al, 2019; DOI: 10.3390/jcm8071027]. However, we did not find any literature indicating that leptin is specifically secreted through exosomes in the context of HCC. To clarify this, we have not included leptin specifically in Figure 3 as we do not feel it would be accurately representing its contribution in this context.
The effect of adiponectin to stimulate the biogenesis of exosomes by adipocyte is also missing in the figure.
We have included this figure 3.
The stimulation of the inflammation is not mentioned in the figure, while it is well described in the text for both AdExo and HCCexo. Idem for cell proliferation and spreading.
We want to thank the reviewer for pointing the important role of inflammation in both the hypertrophic adipocyte and the cancer cell microenvironment and its role in the promotion of carcinogenesis We have included this in figure 3.
Generally, the pathways affected by AdExo and HCCexo and mentioned in Figure 3 are a mixed of specific pathway (eg: MMP2-9, or PI3K/AKT) or more generic/global cellular event (eg: angiogenesis, EMT). It may help if the authors could separate the 2 levels and show the impact of specific pathways on generic/global cellular event when possible.
This is an excellent observation. To further distinguish more specific pathways from global cellular events we have utilized different arrow colors.
The zoom in AdExo show that these exosomes contain RNA, DNA, Metabolites, lipids etc, yet in the text and Table 1 all these elements are not mentioned (eg DNA or metabolites). Either the authors expand on these molecules missing in the text and their role in AdExo or HCC exosomes in the context of HCC, either the list of elements present in these exosomes should be reduced. Could the molecules frequently observed across studies cited in the text and Table 1 be named in figure 3 (eg miRNA148b, miRNA4269, miRNA23b, miRNA4269, etc)? Would it be possible to specify the content of HCCexo as well in the figure? This would highlight the specificity of each exosome subpopulation.
We agree that the zoom of the exosomal contents was reflective of a general exosomal content rather than the specific contents highlighted in this review. As mentioned above, we have modified the contents of both the AdExos and HCCExos accurately reflecting the literature covered in this manuscript.
It would be good if the authors could find a way to highlight the impact on different pathways on adipocytes by the HCCexosomes (I’m guessing it is the list on left side of the graph), and the impact of AdExo on liver cells (I’m guessing it is the list on the right side): either using a colour coding or moving the list for HCCexosomes on the top left of the schema as the HCCexo are on the top of the schema.
As previously mentioned, we have color coded both specific effects and more systemic effects to further clarify the effects of both the microenvironment of adipocytes and the tumor. We have also have added additional elements to the figure to further clarify these effects.
I suggest to the author to revise this figure and make it as complete as possible to reflect their text, as this figure could be a key one for their review that would stick in the mind of any readers and could be cited and mentioned later in conferences etc.
We thank the reviewer for all these recommendations. We truly believe this figure is now more visually appealing and a good representation of the literature covered. Please find all changes mentioned in Figure 3 of revised manuscript.
- Minor: In the introduction define HCC again.
We have defined HCC in the introduction as indicated in line 27.
- Line 151: typo: "ESCORT I" should be "ESCRT I"
We have corrected this typo in section in the revised version of section “4.1. Exosome Biosynthesis”.
- Line 213: “AdExos miRNAs and chronic inflammation:” it seems this is a new subparagraph, but probably not correctly formatted.
Reviewer is correct. AdExos miRNAs and chronic inflammation is now subtitle “5.1 AdExos miRNAs and chronic inflammation” as indicated in line 230.
Reviewer 2 Report
This manuscript is an interesting contribution on the role of exosomes in the crosstalk between adipocytes and liver cancer cells.
Comments:
Some concept are reported as a well-established knowledge, while many aspects regarding the EVs still need to be completely clarified. For example exosome secretion and cell signaling are not really well established (as ISEV suggested: J Extracell Vesicles. 2019 Nov 8;8(1):1684862).
In the text there are some repetition of concepts: for examples in paragraph 3, multiples time has been described that cancer-cell adipocyte crosstalk results in acquisition of fibroblast-like phenotypes (lanes 94 and 113). I would suggest a better organization of the text avoiding repetition of the same concepts.
In the text is used the word exosomes instead of the general EVs. The methodologies currently available don’t allow a definitive separation of exosomes from the EVs. So it is not really completely correct to talk about exosomes but in general it is suggested to use the more general term EVs.
lane 267: the sentence is not finished?
In the table p. 8 two times: miRNANAs?
Author Response
August 20, 2020
Re: The Role of Exosomes in the Crosstalk between Adipocytes and Liver Cancer Cells
Reviewer 2
This manuscript is an interesting contribution on the role of exosomes in the crosstalk between adipocytes and liver cancer cells.
We thank this reviewer for his/her insights and recommendations.
- Some concept are reported as a well-established knowledge, while many aspects regarding the EVs still need to be completely clarified. For example exosome secretion and cell signaling are not really well established (as ISEV suggested: J Extracell Vesicles. 2019 Nov 8;8(1):1684862).
We understand that there is no current technology to excluded all other EVs; however, for the purposes of this publication we have focused on literature that specifically cites exosomes.
We agree with the reviewer that the term “exosome” is broadly used for several forms of EVs; however, we maintain using the term exosome as supported by the various references that we used that specifically engaged that term. The agreed diameter for this subgroup of EVs spans from 30 to 150 nm. We also know that exosomes have different biogenesis than microvesicles and apoptotic bodies as described, for example. Thus, we are confident in using the term “exosome” for this review best identify the group of EVs we are targeting and describing. However, we have added a few sentences and specifically addressed this issue as indicated in lines 123-125 and 142-146. We want to thank this reviewer for bringing this to our attention.
- In the text there are some repetition of concepts: for examples in paragraph 3, multiples time has been described that cancer-cell adipocyte crosstalk results in acquisition of fibroblast-like phenotypes (lanes 94 and 113). I would suggest a better organization of the text avoiding repetition of the same concepts.
Reviewer is correct and we have re-organized section “3: Adipocytes and the tumor microenvironment” to simplify its reading and avoid repetition of concepts. Changes are indicated in lines 83 to 114.
- In the text is used the word exosomes instead of the general EVs. The methodologies currently available don’t allow a definitive separation of exosomes from the EVs. So it is not really completely correct to talk about exosomes but in general it is suggested to use the more general term EVs.
We understand that the terminology for exosome is not entirely correct since more stringent techniques are needed to effectively separate exosomes from other EV’s. Although multiple techniques have been described and recently developed [Yang et al, 2020; DOI:10.7150/thno.41580], there is still the issue of appropriately separating EV’s once they have entered the extracellular space as well pointed out in the findings by Russel et et, 2018 [DOI: 10.1080/20013078.2019.1684862]. We will utilize the terminology exosome since the majority of the publications cited utilize this term. However as indicated below, we choose to specifically address this issue in lines 123-125 and 142-146 of the revised manuscript.
- lane 267: the sentence is not finished?
Thank you for bringing this to our attention. We have re-structured this sentence as indicated in lines 282-284.
- In the table p. 8 two times: miRNANAs?
We have corrected this typo in the table. Changes highlighted in yellow.
Round 2
Reviewer 1 Report
The authors addressed all my comments and improved their review. The figures are nicely designed. The paper can be published as it is.